# Predicting Climate Change Impact on the Habitat Suitability of the *Schistosoma* Intermediate Host *Oncomelania hupensis* in the Yangtze River Economic Belt of China

**DOI:** 10.3390/biology13070480

**Published:** 2024-06-27

**Authors:** Yimiao Li, Mingjia Guo, Jie Jiang, Renlong Dai, Ansa Rebi, Zixuan Shi, Aoping Mao, Jingming Zheng, Jinxing Zhou

**Affiliations:** 1School of Ecology and Nature Conservation, Beijing Forestry University, Beijing 100083, China; liyimiao22@bjfu.edu.cn (Y.L.); maoap0505@bjfu.edu.cn (A.M.); 2School of Soil and Water Conservation, Beijing Forestry University, Beijing 100083, China; gmj0407@bjfu.edu.cn (M.G.); dairl525@bjfu.edu.cn (R.D.); 2013ag3549@uaf.edu.pk (A.R.); shizixuan02@bjfu.edu.cn (Z.S.); 3Schistosomiasis Control Station of Junshan District, Yueyang 414005, China

**Keywords:** climate change, environmental factors, MaxEnt, snail, *Oncomelania hupensis*, schistosomiasis, species distribution modelling

## Abstract

**Simple Summary:**

*Oncomelania hupensis* is the exclusive intermediary host of *Schistosoma japonicum* in China. The alteration of the *O. hupensis* habitat and population distribution directly affects the safety of millions of individuals residing in the middle and lower regions of the Yangtze River, as well as the ecological stability of the Yangtze River Basin. In this study, we used species distribution modeling (SDM) methods to predict the potential distribution of *O. hupensis* in the Yangtze River Economic Belt and found that with climate change, the distribution area would decrease in size and distribution center would shift northward in the future.

**Abstract:**

*Oncomelania hupensis* is the exclusive intermediary host of *Schistosoma japonicum* in China. The alteration of *O. hupensis* habitat and population distribution directly affects the safety of millions of individuals residing in the Yangtze River Economic Belt (YREB) and the ecological stability of Yangtze River Basin. Therefore, it is crucial to analyze the influence of climate change on the distribution of *O. hupensis* in order to achieve accurate control over its population. This study utilized the MaxEnt model to forecast possible snail habitats by utilizing snail distribution data obtained from historical literature. The following outcomes were achieved: The primary ecological factors influencing the distribution of *O. hupensis* are elevation, minimum temperature of the coldest month, and precipitation of wettest month. Furthermore, future climate scenarios indicate a decrease in the distribution area and a northward shift of the distribution center for *O. hupensis*; specifically, those in the upstream will move northeast, while those in the midstream and downstream will move northwest. These changes in suitable habitat area, the average migration distance of distribution centers across different climate scenarios, time periods, and sub-basins within the YREB, result in uncertainty. This study offers theoretical justification for the prevention and control of *O. hupensis* along the YREB.

## 1. Introduction

Schistosomiasis, commonly referred to as “big belly disease”, is an infectious or parasitic ailment that arises when people or mammals, such as cattle, sheep, and pigs, become infected with *Schistosoma japonicum* [1]. Based on the 2020 data from the World Health Organization, schistosomiasis is predominantly prevalent in sub-Saharan Africa and affects approximately 230 million individuals worldwide [2]. China was previously among the nations with the most severe transmission of schistosomiasis globally, and the disease primarily occurs in the Yangtze River basin inside China [3]. A comprehensive survey undertaken in the mid-1950s revealed that schistosomiasis was widespread in the Yangtze River region and 12 provinces located to the south [4]. The survey also indicated that the number of affected individuals exceeded 10 million [4]. Currently, China has a low prevalence of schistosomiasis [5], and considerable progress has been made in the prevention and control of schistosomiasis. However, in recent years, there has been an expansion of snail habitats due to various factors including development along the Yangtze River Economic Belt, wetland ecological protection, aquaculture, land use, and climate change. As a result, the transmission of schistosomiasis has intensified in certain areas where the disease is prevalent [6].

In the regions of China affected by schistosomiasis, *Oncomelania hupensis*, an amphibious freshwater organism, is the sole intermediate host of *S. japonicum* [7]. Snail distribution dynamics can serve as an early warning indicator for the resurgence of schistosomiasis [8]. Hence, regulating the reproduction and dispersal of the snail is a paramount approach in the management of schistosomiasis [9]. There are three types of mollusciciding: physical, chemical, and biological [7]. Nevertheless, the enactment of the “Law of the People’s Republic of China on the Protection of the Yangtze River” in 2021 has brought about significant challenges for conventional methods of snail (*O. hupensis*) and schistosomiasis control, such as the use of chemical molluscicides and the modification of habitats, due to their potential harm to the wetland ecology [10,11]. The law places a strong emphasis on the protection and restoration of the Yangtze River’s ecological environment. Hence, the accurate anticipation and management of snail dispersal are pivotal actions to mitigate environmental hazards.

The distribution patterns of *O. hupensis* are significantly influenced by global climate change. According to the Intergovernmental Panel on Climate Change (IPCC), global warming is projected to reach a minimum of 1.5 °C, resulting in substantial changes to the strength, occurrence, and length of extreme weather occurrences [12]. In light of global warming, forecasting alterations in species distribution patterns as a result of climate change has emerged as a significant field of ecological investigation in China, particularly in the upper regions of the Yangtze River, as this area is particularly susceptible and responsive to climate change in China [13]. The Coupled Model Intercomparison Project (CMIP) is commonly employed for projecting future climate change. Additionally, numerous studies have demonstrated that climate change, including high precipitation, extreme droughts, and warming, directly affects the spread of *O. hupensis* [14,15,16,17]. Hence, it is crucial to forecast the forthcoming alterations in the spatial distribution of *O. hupensis* in diverse climate scenarios to effectively prevent and manage the danger of schistosomiasis in the Yangtze River Basin and for enriching the database on changes in species patterns.

Ecological niche modeling is frequently used to analyze the range, trend, and area of species, it has become a popular method for predicting how species distribution may alter due to climate change [18,19,20]. A suitable predictive model for snail population distribution can effectively forecast the spread of *S. japonicum* [21], thus facilitating precise snail control and minimizing the ecological risk in the Yangtze River Economic Zone. Maximum Entropy Modelling (MaxEnt) is a species distribution model that uses species distribution data and environmental variables to estimate the possible distribution of species in a specific area [22,23]. The MaxEnt model exhibits a strong predictive capacity and is extensively employed for simulating the potential distribution ranges of species [20]. Research on the determinants of snail populations and their management strategies has revealed that environmental factors, including temperature, precipitation, elevation, and sun radiation, can impact the development and breeding of *O. hupensis* [24,25,26,27]. Consequently, these parameters can serve as predictors for the snail’s distribution, thereby simplifying the utilization of the MaxEnt model to forecast appropriate habitats and changes in distribution due to global climate change [28].

This work uses the historical occurrence data of *O. hupensis* and applies the MaxEnt model together with ecological predictive parameters to create an environmental suitability model for the snail. The model is then used to map the probable regional distribution of the snail under various climate scenarios. Following that, it examines the changes in the locations of the *O. hupensis’* distribution centers, intending to answer the following important inquiries: (1) What are the primary environmental elements that influence the distribution of *O. hupensis*? (2) What is the impact of various climate scenarios on the distribution of snails in the Yangtze River Economic Belt? This study is the inaugural attempt to integrate projections of the spatial distribution of *O. hupensis* in the Yangtze River Economic Belt with the phenomenon of climate change. The results will furnish theoretical substantiation and empirical evidence for the prevention and management of *O. hupensis* in the Yangtze River Economic Belt. Additionally, they will provide a logical foundation for the allocation of government resources in disease prevention and control, as well as the formulation of policies for snail population management.

## 2. Materials and Methods

### 2.1. Study Area

The Yangtze River Economic Belt (YREB) is the region in China with the highest prevalence of schistosomiasis. The Yangtze River, the lengthiest river in China and the third lengthiest globally, possesses a primary channel that extends around 6400 km [29]. Most of the basin has a subtropical monsoon climate, with an average annual temperature of about 13.0 °C and an average annual precipitation of about 1067 mm, which is unevenly distributed within the year, with more than 60 percent of the precipitation concentrated in the summer months (June–August) [30]. It is the home to two species of freshwater cetaceans, 424 species of fish, more than 1200 species (genera) of phytoplankton, 753 species (genera) of zooplankton, 1008 species (genera) of benthic animals, and more than 1000 species of higher aquatic plants [31]. The Yangtze River basin spans a large region where the primary channel and its tributaries flow, it is situated between 90°30′–122°25′ E and 24°30′–35°45′ N, encompassing a total area of 1.8 million km^2^ [32]. This area represents 18.8% of China’s whole land area. The YREB encompasses 11 provinces and cities, namely Shanghai, Zhejiang, Jiangsu, Anhui, Jiangxi, Hubei, Hunan, Chongqing, Sichuan, Yunnan, and Guizhou, and it spans a total area of around 2.05 million square kilometers [33]. The upstream of the Yangtze River traverses the provinces of Sichuan, Yunnan, Guizhou, and Chongqing. The midstream passes through Hubei, Hunan, and Jiangxi. The downstream extends through Anhui, Jiangsu, Zhejiang, and Shanghai. This region not only occupies a crucial strategic location in China’s economic development but also garners significant interest due to its ecological security [34].

### 2.2. Data Acquisition

#### 2.2.1. Occurrence Data for *Oncomelania hupensis*

This paper employed a literature review method to gather data on the specific locations where the *O. hupensis* is found. The data were extracted from the published literature in both English and Chinese, obtained from Google Scholar (https://scholar.google.com/ (accessed on 7 September 2023)) and China National Knowledge Infrastructure (https://www.cnki.net/ (accessed on 11 December 2023)). Additional data were supplemented from the Global Atlas of Parasitic Infections (GAHI) (http://www.thiswormyworld.org/ (accessed on 10 September 2023)). The specified search term was “Chinese snails” without any defined time range. Criteria for inclusion in this study were: (1) the snail sampling location was in the Yangtze River Economic Zone; (2) the survey time was specified, at least up to a certain year; (3) the survey locations were specified to the level of villages or floodplain and smaller units; (4) the article had to be an original research article, not a review article; (5) the full text of the article was required. Any papers that did not meet these conditions were excluded [35]. Google Maps (https://www.google.com/maps/ (accessed on 30 December 2023)) was utilized to enhance the coordinate information of sample points by associating them with place name records. Points with incomplete distribution information and unlocatable coordinates were excluded. The geographic data of the obtained distribution points were imported into ArcGIS 10.8 software for verification. Ultimately, a total of 367 valid *O. hupensis* distribution points were identified within the study area. These points span from 1972 to 2022 and were recorded at the village level. They all represent natural distribution sites (Figure 1, Appendix A).

#### 2.2.2. Environmental Data

The environmental factor data used in this study were obtained from the WorldClim database (http://www.worldclim.org/ (accessed on 15 January 2024)). The data have a spatial resolution of 2.5 min per layer.

##### Environmental Data of Snail Current Distribution

This article examines the impact of 19 bioclimatic parameters, solar radiation from January to December, and elevation on the distribution of species. These environmental variables were chosen under the current climate scenario (1970–2000, from wc 2.1) and are presented in Appendix A. Using ArcGIS 10.8, all environmental layers were masked based on the administrative boundaries of the YREB. Subsequently, the geographic coordinate system of each environmental layer was unified as GCS_WGS_1984, and finally, the environmental layers were unified and exported to ASCII format for MaxEnt modeling operations using the Extract by Mask tool in the SDM toolbox 2.5 to establish the initial model.

To mitigate the influence of multicollinearity among predictor variables on the accuracy of model predictions, we computed the correlations between environmental variables using the Raster Correlations and Summary Statistics tool available in the SDM toolbox. Variables with an absolute correlation coefficient of less than 0.8 and a contribution rate of more than 1% to the initial model were retained.

##### Future Climatic Data

The screened bioclimatic factors were selected from the Beijing Climate Center Climate System Model (BCC-CSM2-MR) as part of the Global Coupled Model Intercomparison Project Phase 6 (CMIP6), to serve as the climatic data for the prediction of the future distribution of *Oncomelania hupensis* (all the parameters of the model were improved compared with the BCC-CSM 1.1 model in the CMIP5, which can better simulate the climate distribution [36] and applies to China’s climate change [37,38]). Four shared socioeconomic paths (SSPs) under the model were selected: SSP1-2.6 (sustainability with low-level greenhouse gas emissions), SSP2-4.5 (middle of the road with middling greenhouse gas emissions), SSP3-7.0 (regional rivalry with high–mid-level greenhouse gas emissions), and SSP5-8.5 (fossil-fueled development with high-level greenhouse gas emissions), and four future time periods for each scenario were selected: 2021–2040, 2040–2060, 2061–2080, and 2081–2100. In order to maintain the comparability of the model in the time series, elevation, and solar radiation factors were kept constant in modeling the potential future distribution. The environmental layers are treated in the same way.

### 2.3. Methodology

#### 2.3.1. Establishment and Evaluation of the Species Distribution Model for *O. hupensis*

The MaxEnt 3.4.3 program was utilized to make predictions on the likely distribution zones for the species. The data on the distribution points of *O. hupensis* and the environmental layers needed for modeling were loaded into MaxEnt. The software utilized the Jackknife test to examine the primary environmental parameters that impact the dispersion of the snail. Additionally, response curves were generated to assess the appropriateness of habitat conditions for the snail. The predictive performance of each model was evaluated using the Area Under the Receiver Operating Characteristic Curve, based on the specified criteria [39]. An AUC (Area Under the Curve) value between 0.5 and 0.6 suggests very low simulation accuracy and is not suitable for further analysis. A value between 0.6 and 0.7 indicates poor results, but it can still be marginally used. An AUC value between 0.7 and 0.8 suggests average results and is considered usable. A value between 0.8 and 0.9 indicates good results, while an AUC value between 0.9 and 1.0 indicates excellent results. In order to estimate the spatial distribution of species using accurate models, we conducted 10 replicate runs with a checked random seed, while keeping all other parameters at their default settings. In addition to AUC, TSS (True Skill Statistic), Overall Accuracy, Sensitivity, and Specificity were used to assess the model prediction accuracy [40]. The following ranges were used to interpret TSS statistics: values from 0.2 to 0.5 were poor, values from 0.6 to 0.8 were useful, and values larger than 0.8 were good to excellent [41].

#### 2.3.2. Change in Potential Distribution and Centroids

The projected data for future climate scenarios were inputted into the “Projection layers directory/file”, and the calibrated species distribution model was utilized to forecast future species distribution. The methodology employed for the analysis was as follows: First, the Jenks’ natural breaks categorization approach was employed to categorize the existing distribution of *O. hupensis* into four distinct groups: Very low suitability (0–0.10), Low suitability (0.10–0.31), Moderate suitability (0.31–0.62), and High suitability (0.62–1.00). Uniform classification criteria were used to manually divide the snail’s projected distribution zones into segments. In addition to this, for specific practical applications such as conservation management, the predicted probability of occurrence of *O. hupensis* under the present climate conditions was converted to a binary value using 0.5 as a threshold to produce a binary map indicating the presence or absence of snails [42].

Secondly, for each scenario and time period, the distribution of highly suitable areas in the three reaches of the Yangtze River was condensed into a single central point using the ‘Spatial Statistics Tools-Measuring Geographic Distributions-Median Center’ in ArcGIS. The straight-line distance and the annual distance of the distribution centers of *O. hupensis* movement in the three basins in each 20-year-period were calculated according to the formula. In addition [43], this analysis was used to study the changes in future potential distribution areas and the trajectories of distribution centers. Habitat suitability reclassification, distribution center calculation, and mapping were carried out using ArcGIS software.

#### 2.3.3. Regionalization and Analytical Methods

Because the temperature and precipitation of the YREB and other climatic conditions from the upper reaches to the lower reaches vary significantly, the schistosomiasis-infected areas are divided into three major clusters: water network, lake and marsh, and hills, with the first two types of areas referring to the five provinces of Anhui, Hunan, Hubei, Jiangxi, and Jiangsu, and the hill-type infected areas referring to the two provinces of Sichuan and Yunnan; the YREB is artificially divided into three clusters according to the above. The specific clusters are as follows: upstream (Sichuan, Yunnan, Guizhou, and Chongqing Municipalities), midstream (Hubei, Hunan, and Jiangxi Provinces), and downstream (Anhui, Jiangsu, Zhejiang, and Shanghai Municipalities). An analysis was conducted to examine the habitat changes of the snail in the upstream, midstream, and downstream between 2020 and 2100, considering each region separately.

## 3. Results

### 3.1. The Establishment and Effect of the Model

#### 3.1.1. Filtering of Environment Variables

The selected environmental variables for constructing the *O. hupensis* distribution model were determined by combining the correlation coefficients between the 32 environmental variables (Appendix A) and the relative contribution of each variable to the initial model (Appendix A). The selected variables include elevation, minimum temperature of the coldest month, precipitation of the wettest month, precipitation seasonality, and solar radiation in May, July, and October (Table 1).

#### 3.1.2. Evaluation of Models

Appendix A and Table 2 demonstrate that after 10 iterations of the model, the average test set AUC value of the model was determined to be 0.915, with a standard deviation of 0.010. The TSS value also reached 0.688, indicating that the model results were usable. These results indicate a usable simulation accuracy. The MaxEnt model accurately predicts the possible distribution area of *O. hupensis* with high reliability and no unpredictability. This prediction may be utilized to assess snail distribution in the YREB.

### 3.2. The Influence of Environmental Factor

The primary elements determining the species distribution patterns are environmental factors, which collectively account for about 60%. The distribution of *O. hupensis* is primarily influenced by elevation (elev), the minimum temperature of the coldest month (bio6), the precipitation of the wettest month (bio13), and precipitation seasonality. Among these factors, elevation has the highest contribution rate of 40.1% (Table 1). In addition, the results of the Jackknife test indicate that when elevation is employed as the only variable, it achieves the highest values for training gain, test gain, and AUC. In contrast, when elevation is excluded and only other factors are considered, there is a significant fall in training gain, test gain, and AUC values. This suggests that elevation is not only the most useful variable, but it also provides distinct information that other factors lack. Elevation plays a critical role in limiting the dispersion of the snail (Figure 2).

The response curves for each environmental variable demonstrate the impact on the MaxEnt predictions, revealing how the likelihood of presence fluctuates with changes in each environmental variable while keeping other variables at their average values. It is well known that when the probability of distribution exceeds 0.5, the corresponding ecological factor value is considered favorable for species expansion [44]. The findings indicated that an elevation range of 0–50 m is suitable for the survival of *O. hupensis*. Additionally, a precipitation level of 70–130 mm during the wettest month and a variation in the minimum temperature of the coldest month between −32–17 °C had a predictive value of less than 0.1 for the probability of the existence of *O. hupensis*. Furthermore, the relationship between elevation and the minimum temperature of the coldest month resulted in a single-peaked curve, as shown in Figure 3.

Furthermore, the acceptable range for solar radiation in July was found to be between 12,400–15,200 kJ m^−2^d^−1^. Similarly, the acceptable range for precipitation seasonally was determined to be 98–116 mm. Additionally, the projected values for solar radiation in both May and October were below 0.1 (Appendix A).

### 3.3. Current Prediction of O. hupensis Distribution

A study was performed to analyze the distribution range and region of *O. hupensis* under the present climate conditions. Figure 4 and Figure 5 illustrate the distribution of the snail under the current climate scenario. The snail is primarily found in northwestern Yunnan Province and eastern Sichuan and densely populates the middle and lower regions of the Yangtze River Basin, including Poyang Lake, Dongting Lake, the Jianghan Plain, tributaries of the Yangtze River in Anhui Province and Shanghai. The habitat areas for each type are as follows: 127.095 × 10^4^ km^2^ (Very low suitability), 32.431 × 10^4^ km^2^ (Low suitability), 15.733 × 10^4^ km^2^ (Moderate suitability), and 13.816 × 10^4^ km^2^ (High suitability) (Appendix A). The areas most appropriate for *O. hupensis* are vast and primarily located in regions with a strong presence of agriculture and fisheries, as well as substantial concentrations of humans involved in water-related activities.

### 3.4. Future Prediction of O. hupensis Distribution

The predicted spatial distribution of *O. hupensis* in the YREB is illustrated in Figure 6 (Appendix A) for four shared socioeconomic paths. Global warming will greatly decrease the distribution of *O. hupensis* in all types of habitats in the future (Figure 7). This reduction in suitable areas will vary depending on different climates. Specifically, the area of high suitable areas for *O. hupensis* in SSP1-2.5 will be the smallest compared to other habitats, while SSP5-8.5 will have the largest area of high suitability areas. Over time, the high suitability zones in SSP1-2.6 and SSP5-8.5 expanded, with SSP5-8.5 experiencing a particularly significant increase. The area of SSP2-4.5 initially decreased and then increased, but it remained lower than the initial period. The suitable habitat in SSP3-7.0 remained relatively stable without much change. The highly suitable habitat area is primarily found in the eastern part of Sichuan Province and the northern parts of Jiangsu and Anhui Provinces. There is only a small amount of distribution in Hubei Province, which is shifted northward compared to the current climate scenario.

### 3.5. Directional Analysis of O. hupensis’ Distribution Centroid

Overall, there is a prevailing inclination for the distribution center of *O. hupensis* to shift toward the north in the coming years (Figure 8). According to the projected climate conditions, the snail distribution center upstream has shifted to the northeast of its current location, while the midstream and downstream snail distribution centers have moved to the northwest of their current location (Figure 8, Appendix A). Examining the snails’ performances in three distinct river basins under a specific climate scenario demonstrates the following: in the SSP1-2.6 scenario, the average distance that the distribution center migrates is greater in the downstream compared to the upstream and midstream. However, under different climate scenarios, the migration distance is greater upstream. Specifically, in the SSP2-4.5 scenario, the average migration distance of the distribution center is upstream > midstream > downstream. Similarly, in both the SSP3-7.0 and SSP5-8.5 scenarios, the average migration distance is upstream > downstream > midstream. When examining individual basins under different climate scenarios, it is observed that the average migration distance of distribution centers in the upstream and midstream varies significantly. The order of variation, from highest to lowest, is as follows: SSP2-4.5 > SSP5-8.5 > SSP3-7.0 > SSP1-2.6. For the downstream, the greatest change in average migration distance occurs under the SSP5-8.5 scenario. The order of variation, from highest to lowest, is as follows: SSP5-8.5 > SSP1-2.6 > SSP2-4.5 > SSP3-7.0 (Appendix A).

## 4. Discussion

Topography is a well-known factor in the distribution of *O. hupensis*. The study found that elevation had the greatest impact and played a significant role in determining the distribution of *O. hupensis*. Prior research has indicated that *O. hupensis* are primarily found at lower elevations, and Luo Zhihong and Yang Yu et al. specifically identified that the region of Dongting Lake beach with snails is predominantly located at elevations ranging from 24 to 32 m [45,46]. Furthermore, they have observed that both lower and higher elevations of the beach are unfavorable for *O. hupensis* breeding. Research conducted in lakes and marshes revealed that *O. hupensis* had a higher chance of surviving in areas with lower elevations compared to those with higher elevations, the average elevation at which *O. hupensis* was located was 17.72 ± 8.64 m [47,48]. Furthermore, the slope also has an impact on the distribution of *O. hupensis*. For instance, Zhu et al. found that an increase in slope leads to a decrease in the likelihood of *O. hupensis* breeding [49].

Climate has a crucial role in determining the geographic range of most species. Of the climate variables examined, the distribution of *O. hupensis* is mostly influenced by the minimum temperature and maximum precipitation. This study suggests that the precipitation of the wettest month is very appropriate for the *O. hupensis* ranges from around 50 to 140 mm. According to research conducted by Cheng Gong, Li Dan, and other scientists, rainfall has a significant impact on the growth and development of *O. hupensis* [50]. This is because rainfall affects the humidity of the snail’s natural environment. The study found that there is a negative relationship between the amount of rainfall in a year and the density of *O. hupensis*. Both excessive and insufficient rainfall can disrupt the ability of *O. hupensis* to reproduce. The suitability modeling of the habitat of *O. hupensis* is significantly influenced by the minimum temperature of the coldest month. Prior research has revealed that cold temperatures throughout the winter season have a significant impact on the ability of snails to enter hibernation and subsequently engage in reproduction. Tang Yang and colleagues believe that the lowest winter temperature has the most significant influence on *O. hupensis* density [51]. The ideal temperature range for *O. hupensis* is 20 °C to 30 °C, temperatures outside of this range can result in delays or complete halt in the snails’ growth and ability to reproduce [52,53]. The physiological activities of *O. hupensis* can be diminished by lower ambient temperatures. Consequently, accurately predicting the northward journey of the *O. hupensis* relies on the presence of low winter temperatures. In addition to temperature and precipitation, solar radiation also has a substantial impact on the dispersal of *O. hupensis*. The study found that solar radiation in the months of May, July, and October played a substantial role in predicting the distribution of snails. Specifically, it accounted for 20% of the overall forecast accuracy. Tai Hongyu and colleagues conducted a study on the spread of schistosomiasis infections in sheep in Qinghai Province [54]. They discovered a direct relationship between the infection rate of schistosomiasis and solar radiation. This correlation may be attributed to the flourishing of algae, which serve as a primary food source for the *O. hupensis*, under high levels of solar radiation [55]. Liu Wenguo’s research revealed that solar radiation has an indirect effect on the distribution of hosts via influencing soil moisture, in addition to its impact on algae [56].

The study found that the distribution range of *O. hupensis* is expected to decrease under future climate scenarios, with the distribution center shifting to the northeast occurs in the upper basin and towards the northwest in the middle and low basins, pointing towards a common location north of the YREB as a whole. These alterations in the distribution area and range are typical among other comparable species and places. The MaxEnt model was employed by the *Abteilung Ökologie* and other researchers to forecast the possible distribution of the common freshwater snail *Radix balthica* in the face of global change scenarios [57]. The findings indicate that the northern boundary of *Radix balthica*’s range is expected to shift moderately towards the north. Predictions indicate that the covering area of its existing occupancy in regions such as France, western United Kingdom, and southern Germany will significantly decline. Yingxuan Yin and colleagues utilized the MaxEnt model to forecast the prospective range of *Pomacea canaliculata* based on projected climate conditions [58]. The findings demonstrated that precipitation in the warmest quarter and maximum temperature in the coldest months played important roles in the distribution of *P. canaliculata*. Furthermore, due to global warming, it is anticipated that the geographical range of *P. canaliculata* would increase and move toward the north. Except for mollusks, many other species’ distributions are also thought to be changing under the influence of climate change, for example, in China, the habitat area of *Tilia amurensis* is shrinking and gradually fragmenting [59], and in Colombia, birds are projected to lose on average between 33 and 43% of their total range under future climate conditions and up to 18 species may lose their climatically suitable range completely [60].

Chinese scholars have observed the possibility for the snail *O. hupensis* to migrate northward in response to future climate change, according to studies on its range [61,62]. Zhou Xiaonong and his colleagues have conducted comprehensive research on the influence of global warming on the spread of schistosomiasis in China [15,62,63]. They utilized Geographic Information System technology to develop a climate–transmission model, which is based on the cumulative temperature required for the growth of *O. hupensis* and *Schistosoma japonicum*. According to their models, increasing winter temperatures could lead to the expansion of schistosomiasis epidemic areas toward the north. The findings suggest that snail expansion in eastern China is more significant compared to central and western China. This species of snail has the potential to spread northward to provinces such as Shandong, Hebei, Shanxi, and even as far as Xinjiang within the next 50 years [63]. This projected range exceeds the predictions made in this study. However, Zhou Xiaonong and colleagues solely considered temperature as the crucial factor for their forecasts, disregarding the influence of other environmental parameters like elevation. This oversight may have resulted in an overestimation of the projected range. Seasonal flooding results in the destruction of many snail populations [64], so more data are needed on future flooding scenarios that can be used as important factors in niche analyses. This work focuses exclusively on the Yangtze River Basin and does not investigate the possibility of *O. hupensis* spreading to other water systems outside of this region. Further research is needed to explore this topic.

Currently, there is a lack of research regarding the future migration patterns and mechanisms of *O. hupensis*. A study’s findings show that snails tend to move towards regions with higher latitudes and lower altitudes. Upstream migrations are distinct from those in the midstream and downstream. A plausible rationale for the northward movement of the snails is that as temperatures increase as a result of climate change, the temperatures in their original habitats surpass the highest temperature limit for their growth, compelling them to relocate to cooler regions at higher latitudes. The snails in the Yangtze River Basin exhibit a persistent northward movement. However, the specific directions of this migration vary between the upstream, midstream, and downstream sections. This variation can be attributed to variances in regional precipitation patterns. During the summer, when there is a lot of rainfall, the upstream area of the Yangtze River is mainly influenced by the southwest monsoon [65]. On the other hand, the midstream and downstream areas, which are located in eastern China, are primarily affected by the southeast monsoon [66]. In these areas, rainfall often decreases as you move along the direction of the monsoon. In addition, the snails frequently choose appropriate low-lying regions for their survival. The anticipated rise in heavy rainfall due to global warming [67] could potentially cause snails upstream to migrate towards the drier northeast, while snails in the middle and lower parts of the stream may move towards the less rainy northwest. Hence, it is plausible that the observed migration patterns are primarily influenced by the collective impact of temperature, precipitation, and elevation.

The alterations in species distribution within the Yangtze River Basin under future climate change scenarios are uncertain. In this study, the alterations in the appropriate habitat area for *O. hupensis* and the movements in the distance and speed of the distribution center exhibited diverse patterns in different climate scenarios, time periods, and locations. Several research utilizing diverse climate models in CMIP6 have demonstrated that fluctuations in climatic conditions, such as precipitation and temperature [36,67,68,69], manifest distinct patterns across multiple climate scenarios and spatial-temporal scales [59,70]. Hence, combining the alterations in climate variables within the Yangtze River Basin using the BCC-CSM2-MR model with the observed shifts in snail distribution in this study may provide a more comprehensive explanation for this event. Ying Li and colleagues’ research shows that in the 21st century, there is a consistent rise in rainfall in the Yangtze River Basin under both the SSP1-2.6 and SSP5-8.5 scenarios [71]. However, the estimated increase in rainfall is much greater in the SSP5-8.5 scenario compared to the SSP1-2.6 scenario. Increased precipitation stress may be the reason for the heightened activity of snail migration. This study found that the average migration distance and speed of *O. hupensis* in the YREB are greater under SSP5-8.5 compared to SSP1-2.6. Ying Li’s research also contradicts the hypothesis that more rain causes snails to migrate more. The research reveals that the areas with the least amount of rainfall are the lower reaches (lower reaches of the Yangtze River Basin, downstream of Hukou) under SSP1-2.6 and the upper reaches (upper reaches of the Yangtze River Basin, between Yibin and Yichang) under SSP5-8.5. This result contradicts the discovery made in this study that snail migration in the downstream area under SSP1-2.6 and in the upstream area under SSP5-8.5 has higher average distances and speeds compared to other river basins under the same climate scenario. The discrepancy in precipitation conditions and snail migration patterns between different river basins in the Yangtze River Basin may be attributed to the use of different criteria for dividing the regions in the two studies. This complicates the explanation of snail migration dynamics in this study.

The future distribution of the snail is uncertain due to climate change and the complicated geographic and socio-economic conditions of the YREB. This uncertainty poses increased challenges to the livelihoods of people in the region. This study reveals that climate change has worsened the uncertainty in the distribution of the snail, despite a reduction in its future distribution area. The changes in the snail’s distribution area and center under different climatic scenarios, as shown in Figure 6 and Figure 7 and Appendix A, indicate a trend of habitat shifting. This poses a risk to individuals who rely on water for their livelihoods, as they are more susceptible to schistosomiasis. Moreover, research suggests that the chronic diseases caused by schistosomiasis can push families living in low-income rural areas into a “poverty trap” [72]. The projected movement trend of the *O. hupensis* distribution area could jeopardize the government’s efforts to support the destitute populace. Thus, opting for the sustainable green road (SSP1), enhancing the forecasting of potential epidemiological regions following the snail’s northward migration, and establishing a more responsive and effective system of early warning indicators for schistosomiasis monitoring will aid in reducing snail breeding and support the successful execution of national policy. In recent years, research on the application of drones for snail control has been carried out gradually, and studies have shown that drones have the advantages of maneuverability and ease of operation, which provide a new way of applying molluscicides for snail control. However, drones spray a whole area comprehensively according to a predetermined route, thus making the cost of chemical control per unit area higher than that of manual spraying. Therefore, combining model prediction with drones to achieve precise snail prevention and control and avoid ineffective and excessive spraying may be a new direction for future snail research.

## 5. Conclusions

In this study, we utilized the MaxEnt model to forecast possible snail habitats in the Yangtze River Economic Belt of China. Our study found that elevation, minimum temperature of the coldest month, and precipitation of the wettest month are the primary ecological factors that influence the distribution of *O. hupensis*. The combined effect of these factors is expected to result in a reduction in suitable habitats for *O. hupensis* and a shift in the location of distribution centers towards the northeast or northwest. Given the uncertainty surrounding the migration of *O. hupensis* in the context of climate change, it is imperative to make more precise projections and implement stricter regulations regarding snails. These measures align with China’s ecological preservation and poverty reduction programs.

## Figures and Tables

**Figure 1 biology-13-00480-f001:**
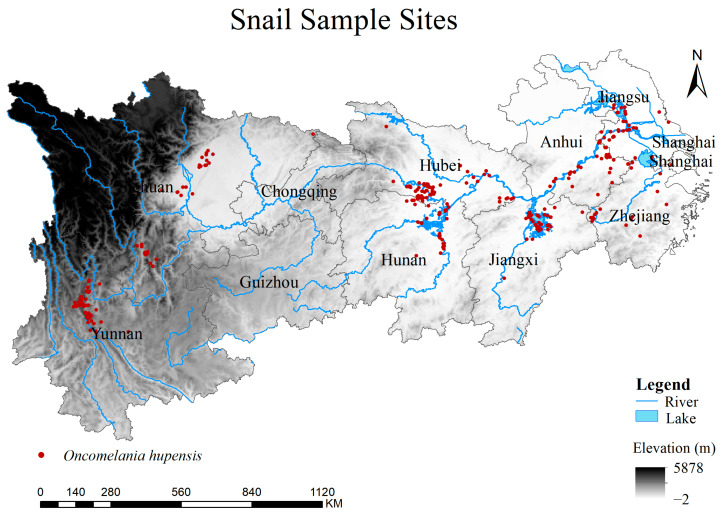
Snail occurrence records of *O. hupensis* in the Yangtze River Economic Belt of China. Red dots represent the *O. hupensis* and the blue lines represent rivers.

**Figure 2 biology-13-00480-f002:**
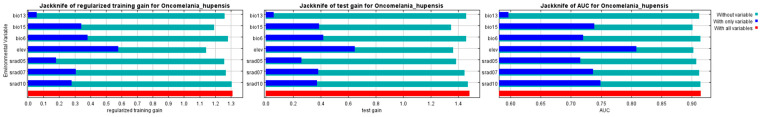
Results of jackknife testing of each variable’s contribution to modeling the *O. hupensis* habitat suitability distribution.

**Figure 3 biology-13-00480-f003:**
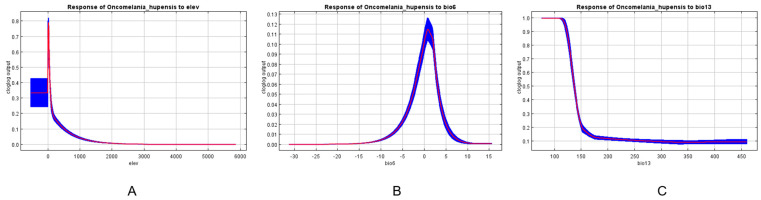
Response curves of MaxEnt models for environmental variables. The red curves represent the average value over 10 replicate runs, while blue margins represent ± SD calculated for 10 replicates: (**A**) elev = Elevation; (**B**) bio6 = Min. Temperature of Coldest Month; (**C**) bio13 = Precipitation of Wettest Month.

**Figure 4 biology-13-00480-f004:**
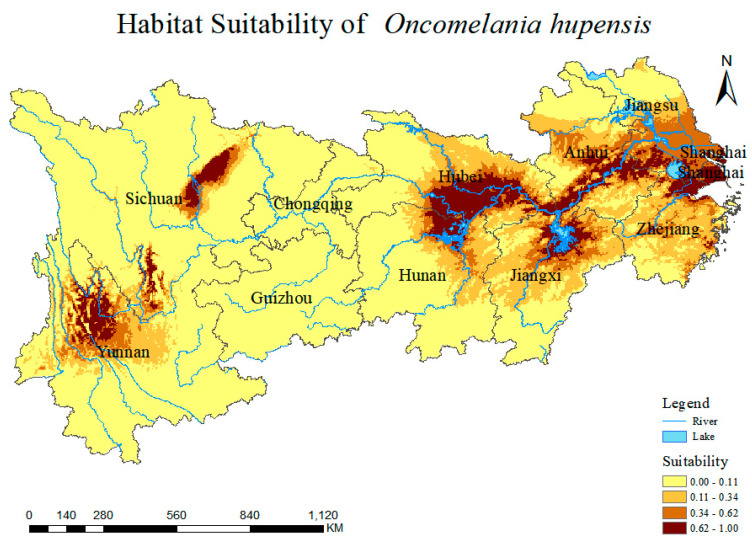
Distribution range predicted by MaxEnt model for *O. hupensis* in the Yangtze River Economic Belt. Habitat suitability classes include Very low suitability (0–0.11), Low suitability (0.11–0.34), Moderate suitability (0.34–0.62), and High suitability (0.62–1).

**Figure 5 biology-13-00480-f005:**
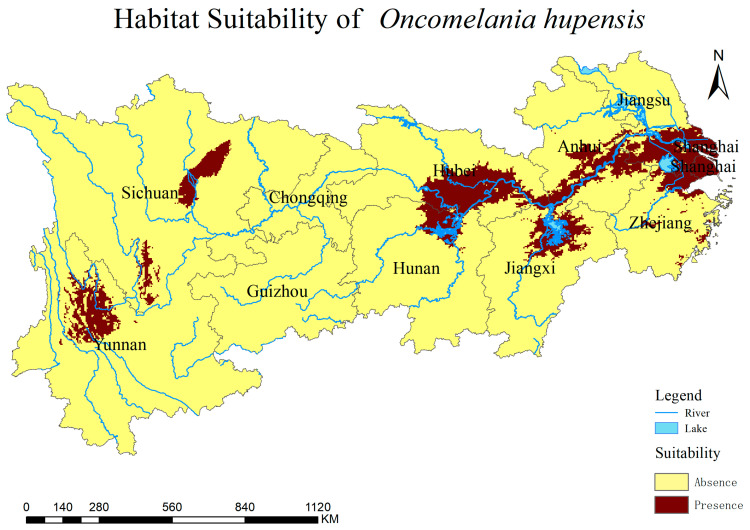
A binary map of *O. hupensis* habitat predicting risk in the Yangtze River Economic Belt under the present climate conditions. Yellow means *O. hupensis* are absent in the area, and reddish-brown means *O. hupensis* is present in the area.

**Figure 6 biology-13-00480-f006:**
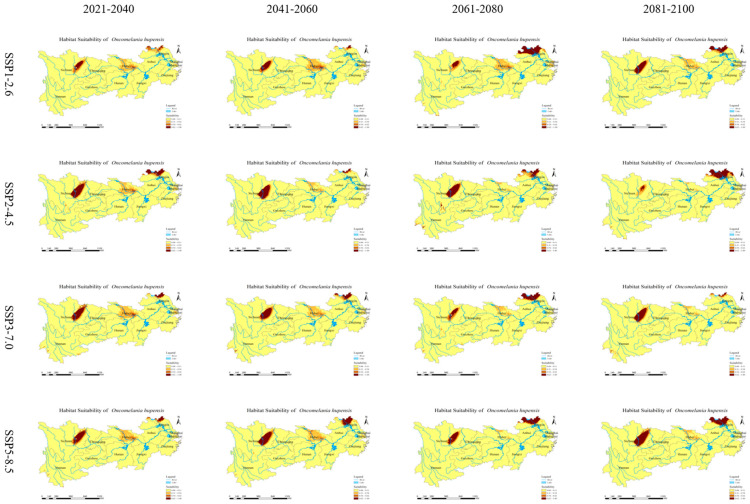
Spatial distribution for *O. hupensis* in the Yangtze River Economic Belt under four shared socioeconomic pathways (SSPs). Habitat suitability classes include Very low suitability (0–0.11), Low suitability (0.11–0.34), Moderate suitability (0.34–0.62), and High suitability (0.62–1).

**Figure 7 biology-13-00480-f007:**
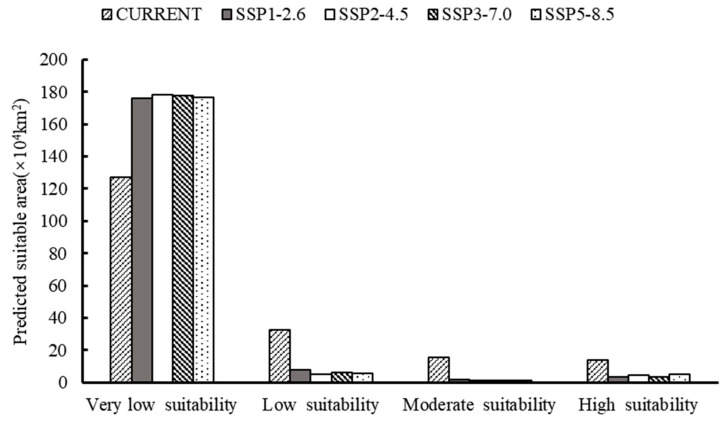
The mean area of suitable areas of *O. hupensis* under current and future climate conditions, as predicted by MaxEnt based on BCC-CSM2-MR climate model.

**Figure 8 biology-13-00480-f008:**
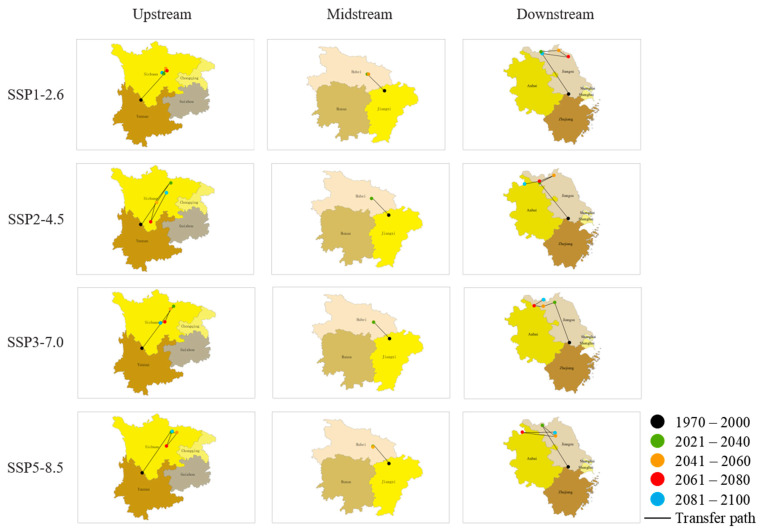
The core distribution shifts under four shared socioeconomic pathways (SSPs) for *O. hupensis* from 2021 to 2100. Transfer path indicates the migration path of *O. hupensis*.

**Table 1 biology-13-00480-t001:** Percent contribution and permutation importance of the environmental variables in the MaxEnt model.

Code	Environmental Variable	Percent Contribution	Permutation Importance
elev	Elevation	40.1	46.8
bio6	Min. Temperature of Coldest Month	18.1	13.3
bio13	Precipitation of Wettest Month	11.9	6.5
srad07	Solar radiation in July	11.3	17.9
srad05	Solar radiation in May	8.7	7.9
bio15	Precipitation Seasonality	8.0	6.3
srad10	Solar radiation in October	1.9	1.4

**Table 2 biology-13-00480-t002:** Model performance of *Oncomelania hupensis* habitat suitability.

Accuracy Measure	Result
AUC	0.915
TSS	0.688
Overall Accuracy	0.898
Sensitivity	0.790
Specificity	0.898

## Data Availability

Data are contained within the article.

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
