# Peer review of "Predicting Climate Change Impact on the Habitat Suitability of the Schistosoma Intermediate Host Oncomelania hupensis in the Yangtze River Economic Belt of China"

_biology, 2024, doi:10.3390/biology13070480_

Round 1
Reviewer 1 Report
Comments and Suggestions for Authors
In the paper “Predicting Climate Change Impact on the Habitat Suitability of Schistosoma Intermediate Host Oncomelania Hupensis in the Yangtze River Economic Belt of China Using MaxEnt” the authors utilised the MaxENT model to forecast possible snail habitats by utilising snail distribution data obtained from historical literature. This manuscript is well organized, and the drawn conclusions are coherent with the obtained results. Despite the fact that I have enjoyed reading your work, I feel that it needs to be corrected by a native English speaker because I have seen a few grammatical errors. I hope to provide very useful suggestions to improve the overall clarity of your study as well as the quality of your analysis. I think that my suggestions look feasible to you, and I believe you will be able to address them. Thus, please take care to do a full revision of your manuscript according to all my comments. Improvements based on my comments will be crucial for acceptance. I have some concerns and suggestions for each aspect of the manuscript. Please see below.
Lines 36 – 36: Please arrange the keywords alphabetically.
Lines 39 – 83: Please, reduce this part.
Lines 88 – 91: I think that you should add these important references to support your sentence: “The Maximum Entropy Modelling (MaxEnt) is a species distribution model that uses species distribution data and environmental variables to estimate the possible distribution of species in a specific area”. I would like to suggest:
Bosso, L., et al., (2024). Integrating citizen science and spatial ecology to inform management and conservation of the Italian seahorses. Ecological Informatics, 79, 102402.
Alaniz, A. J., et al., (2024). Unravelling the cavity‐nesting network at large spatial scales: The biogeographic role of woodpeckers as ecosystem engineers. Journal of Biogeography, 51(4), 710-724.
Lines 115 – 127: I think that you should add some important information also on the biological and climatic conditions of your study area.
Line 145: Did you analyse your data for spatial autocorrelation? From the figure 1 I see that your data were well clustered. What do you think?
Line 154: Why did you choose this spatial resolution?
Line 184 – 199: Please, add the information on all the setting that you selected during the maxent ran. Furthermore, the AUC is not sufficient to assess the performance of your models. Please, use also the TSS method.
Allouche, O., et al. (2006). Assessing the accuracy of species distribution models: prevalence, kappa and the true skill statistic (TSS). Journal of applied ecology, 43(6), 1223-1232.
Line 148: Please, add the north symbol in the map.
Line 118: How did you calculate these three clusters?
Lines 242: Please, move this figure in the supplementary materials.
Line 293: I think that you should show also the binary maps (presence/absence) of your Maxent model. Please, convert the logistic map in the figure f in a binary map by using, for example, the TSS method.
Line 311: These maps were very difficult to view. Please, enlarge these maps.
Lines 340 – 476: The paper discussed appropriately the context and the theme, although there is important literature not cited by the authors. I think that the authors should be discussing their results also comparing them with those already published on other species/genus/family. In fact your paper discusses findings in relation to some of the work in the field but ignores other important work that I think should be added in your discussion considering also other factor of niche analysis.
Lines 439 – 440: I think that you should add these important references to support your sentence: “manifest distinct patterns across multiple climate scenarios and spatial-temporal scales”. I would like to suggest:
Di Febbraro, M., et al., (2023). Different facets of the same niche: Integrating citizen science and scientific survey data to predict biological invasion risk under multiple global change drivers. Global Change Biology, 29(19), 5509-5523.
Chen, B., et al., (2023). Distribution change and protected area planning of Tilia amurensis in China: A study of integrating the climate change and present habitat landscape pattern. Global Ecology and Conservation, 43, e02438.Lines 470 – 476: Please, add some sentences about the future research development.
Lines 466 – 476: Please, add some sentences about the future research development.
Comments on the Quality of English LanguageDespite the fact that I have enjoyed reading your work, I feel that it needs to be corrected by a native English speaker because I have seen a few grammatical errors.
Reviewer 2 Report
Comments and Suggestions for Authors
See attached.

Comments on the Quality of English LanguageEnglish language presentation is fairly good, but requires editing throughout.
Reviewer 3 Report
Comments and Suggestions for Authors
Dear authors, please correct the latine name of species O. Hupensis into O. hupensis in the main text and in figure captions.
Line 259, Figure 3: Please, expand resolution and improve the quality of the figure.
Line 277, Figure 4: Please, expand resolution and improve the quality of the figure.
Line 311, Figure 6: Here, you should provide maps of changes for each scenario separately, because it seems unreadable in the presented form.
Round 2
Reviewer 1 Report
Comments and Suggestions for Authors
Well done!
Comments on the Quality of English LanguageMinor editing of English language required
Author Response
Response to Comments on the Quality of English Language
Point 1: Minor editing of English language required.
Response 1: Thanks for your selfless guidance, a native English speaker partner has been commissioned to conduct a second examination of the manuscript.
Reviewer 2 Report
Comments and Suggestions for Authors
General Comments
The authors made significant changes to the manuscript and addressed most of my comments. Original comments that I feel were not adequately addressed and several lnew comments are provided below.
New Comments
· Abstract, line 38. “MaxEntSnail” should be split into “MaxEnt” and “Snail.”
· Section 1, line 66. Change “medicinal” to “chemical.”
· Section 3.3, Figure 5. This is a new figure with little explanation. Is this predicted presence and absence or current?
· Section 4, line 529 and 531. Change “a new way of applying drugs” to “a new way of applying molluscicides” and “cost of medication” to “cost of chemical control.”
Comments not Adequately Addressed
· Original comment: Section 1, line 58. Which organism, O. hupensis or S. japonicum?
My response: A change was made, but the rewording is problematic. Change to “In the regions of China affected by schistosomiasis, Oncomelania hupensis, an amphibious freshwater organism, is the sole intermediate host of S. japonicum[7].”
· Original comment: Section 1, line 65. What are “snail control forests?”
My response: A change was made, but this is still odd wording. How about changing “the establishment of snail (O. hupensis) control and schistosomiasis prevention forest,” to “the modification of habitats to control (O. hupensis) and schistosomiasis . . .”
· Original comment: Section 2.2.2, lines 177-178. Provide a very brief description of each SSP and their associated radiative forcing levels.
My response: Descriptions were added, but a close parenthesis symbol is missing after description of SSP2-4.5 and an open parenthesis symbol is missing after SSP3-7.0.
· Original comment: Section 3.1.2, line 236. Change “modellings” to “models.”
My response: A change was made, but “models” was misspelled.
· Original comment: Section 3.3, lines 288-289. These lines present unsuitable, low suitability, moderate suitability, and high suitability categories whereas elsewhere the categories are low, general, moderate, and high suitability. See previous comments on these categories.
My response: My original comment was addressed, but “suitable” should be changed to “suitability” for these categories throughout.
· Original comment: Section 3.4, Figure 8. This figure is unreadable due to low resolution. The inset doesn’t seem to add any additional information and could be removed to provide additional space for the maps.
My response: Resolution has been improved, but the maps and legends are unreadable and the inset of trajectories is unnecessary.
Comments on the Quality of English LanguageThere are still some portions of the ms that could use editing for proper English usage.
Author Response
Thank you for your careful guidance, which has greatly contributed to the readability of the article. The following is my reply to your comments:
Comments 1: Abstract, line 38. “MaxEntSnail” should be split into “MaxEnt” and “Snail.”
Response 1: Thank you for pointing this out. We agree with this comment. See lines 38
Comments 2: Section 1, line 66. Change “medicinal” to “chemical.”
Response 2: Agree. See lines 65.
Comments 3: Section 3.3, Figure 5. This is a new figure with little explanation. Is this predicted presence and absence or current?
Response 3: Figure 5 is a binary map of O. hupensis habitat predicting risk in the Yangtze River Economic Belt under the present climate conditions. We have added the complete explanation of the picture in line 317,318 in the blue section, in addition, lines 222-224 can see the purpose and method of making this picture.
Comments 4: Section 4, line 529 and 531. Change “a new way of applying drugs” to “a new way of applying molluscicides” and “cost of medication” to “cost of chemical control.”
Response 4: Agree. See lines 510 and 512.
Comments 5: Original comment: Section 1, line 58. Which organism, O. hupensis or S. japonicum?
Round 2: A change was made, but the rewording is problematic. Change to “In the regions of China affected by schistosomiasis, Oncomelania hupensis, an amphibious freshwater organism, is the sole intermediate host of S. japonicum[7].”
Response 5: Agree. See lines 56 and 59.
Comments 6: Original comment: Section 1, line 65. What are “snail control forests?”
Round 2: A change was made, but this is still odd wording. How about changing “the establishment of snail (O. hupensis) control and schistosomiasis prevention forest,” to “the modification of habitats to control (O. hupensis) and schistosomiasis . . .”
Response 6: Agree. See lines 65 and 66.
Comments 7: Original comment: Section 2.2.2, lines 177-178. Provide a very brief description of each SSP and their associated radiative forcing levels.
Round 2: Descriptions were added, but a close parenthesis symbol is missing after description of SSP2-4.5 and an open parenthesis symbol is missing after SSP3-7.0.
Response 7: Agree. See lines 184 - 187.
Comments 8: Original comment: Section 3.1.2, line 236. Change “modellings” to “models.”
Round 2: A change was made, but “models” was misspelled.
Response 8: Agree. See lines 256.
Comments 9: Original comment: Section 3.3, lines 288-289. These lines present unsuitable, low suitability, moderate suitability, and high suitability categories whereas elsewhere the categories are low, general, moderate, and high suitability. See previous comments on these categories.
Round 2: My original comment was addressed, but “suitable” should be changed to “suitability” for these categories throughout.
Response 9: Agree. See lines 307, 308, 313-315, 335-337,. The pictures and other parts of the manuscript have also been altered(Figure 7, Figure S3).
Comments 10: Original comment: Section 3.4, Figure 8. This figure is unreadable due to low resolution. The inset doesn’t seem to add any additional information and could be removed to provide additional space for the maps.
Round 2: Resolution has been improved, but the maps and legends are unreadable and the inset of trajectories is unnecessary.
Response 10: The trajectories has been removed and the legend is placed separately in the main diagram. See Figure 8.